# Improving Equitable Access to Graduate Education by Reducing Barriers to Minoritized Student Success

David M. Rehfeld [1,*], Rachel Renbarger [2], Tracey Sulak [3], Abby Kugler [1] and Payton DeMeyer [1]

1   Department of Applied Human Sciences, Kansas State University, Manhattan, KS 66506, USA
2   Fhi360, Durham, NC 27701, USA; rrenbarger@fhi360.org
3   Department of Educational Psychology, Baylor University, Waco, TX 76798, USA; tracey_sulak@baylor.edu
*   Correspondence: drehfeld@ksu.edu

**Abstract:** Structural inequities in graduate education perpetuate inequity for students with historically minoritized identities. This paper reviews previous reports of inequities faced by students with minoritized identities and suggests a path forward for improving equitable access to doctoral study. Specifically, this paper suggests investing in the scholarship of teaching and learning while using Gardner's model of doctoral student development to provide targeted support at different levels of operation: the institution, the department, and the individual. Evidence for suggested supports is also provided and a call for further research on the effects of such programs for recruitment, retention, and graduation of minoritized students is made.

**Keywords:** doctoral student development; equity in higher education; scholarship of teaching and learning





## 1. Introduction

Graduate education suffers from large disparities in doctoral students' access to, and ultimate success in, PhD programs related to systemic issues of racism, classism, homophobia, and sexism [1]. These issues disproportionately impact historically minoritized communities and are perpetuated by existing systems' preference for the status quo. Although the COVID-19 pandemic spurred universities to reconsider the academic and social–emotional supports available to students, access to counseling and related services does little to address challenges within a larger system that has historically viewed success from the perspective of the privilege that stems from membership in a majority culture. If universities are serious about promoting equity as more than a token consideration in their graduate education programming, the field must consider the ways in which existing systems are designed to promote not only a narrow view of success but also an even narrower road to achieve it that may be strongly influenced by students' identification with one or more privileged identities. As such, this paper conceptually highlights several identity groups that have been disproportionately impacted by systemic barriers to their success in higher education. The inclusion of these identity groups (race and ethnicity, socioeconomic class and background, and gender identity) does not suggest that other identities do not face systemic barriers to their success in higher education, but rather is intended as a series of examples on how systemic barriers might manifest for different groups and how these barriers might be mitigated. It is also important to note that this conceptual work is written within the context of higher education as it currently exists in the United States, which directly influences the identities chosen as examples as well as the concepts and terminology used in this work. Readers are encouraged to reflect on the history and systems in which they operate to determine how these examples and recommendations translate into their own contexts.

The scholarship of teaching and learning (SOTL) is an evidence-based approach to higher education that has emerged as a field over the past 30 years [2]. Although it is

often conceptualized at the level of individual instructors or classrooms, it can also be implemented at broader levels spanning from individual departments to entire fields of study [3]. In this way, SOTL may help serve as an implementation and dissemination framework within which programs, departments, colleges, and universities can reconsider the influence of existing systems in perpetuating the ongoing marginalization of students from non-majority cultures. A critical feature of SOTL work is that it is both systematic in its review of available evidence as well as responsive to what that evidence suggests, especially when considering that teaching goes beyond the classroom and also includes movement toward desired student outcomes [4]. Accordingly, when available evidence suggests that certain groups experience barriers to their success that may be directly related to their identification or membership in such groups that should, in theory, be irrelevant to their capacity for success, a SOTL approach would hold that educators should reflect on changes that would help to facilitate student success both within and outside of the physical classroom. Accordingly, the concept of a "hidden curriculum" is referenced throughout this work to underscore the idea that the field of higher education often operates within a cultural framework in which certain assumptions are held implicitly but rarely made explicit. As such, this advantages people with privileged access to shared cultural identities while disadvantaging people from historically marginalized identity groups.

### 1.1. Race and Ethnicity

Previous research has documented that students' racial and ethnic identities influence their perceptions of campus climate, especially when that cultural identity is not reflected in the majority culture at an institution. For example, Briscoe reported that Black graduate students are often navigating the aftermath of traumatic events on university campuses like their non-minoritized peers but also bear the additional burden of having to navigate institutional responses that fail to acknowledge the reality of minoritized students' lived experiences on predominantly white campuses [5]. Similarly, students of color frequently report being disproportionately "asked" to engage in service-related activities at a rate that is inconsistent with expectations for long-term success related to tenure and promotion within academia [6]. Minoritized racial identities have also been associated with "othering" and are linked to having a limited connection to institutions and curricula [7,8]. At its core, previous research suggests that students of color are tasked with doing more by systems that exclude them and encourage their success less. Kaler-Jones and colleagues reported on the experiences of students of color and their perceptions of pedagogical practices that facilitated their inclusion and engagement in a hybrid graduate program [9]. Specifically, these students reported that faculty who specifically created space for conversations related to social justice and current events, fostered classroom dialogue and discussions, and encouraged collaboration both within and outside of the classroom facilitated their engagement with the curriculum and created a space in which they felt more meaningfully included.

### 1.2. Socioeconomic Class and Background

Previous research also suggests that students who come from lower socioeconomic backgrounds are minoritized by existing systems in higher education. Students whose families operate with lower levels of socioeconomic capital cannot depend on generational wealth to enable the majority of their focus to be on their studies. This also means that these students likely need to work part- or full-time during their program of study, potentially in addition to taking out larger amounts of loans to fund their education [10]. Financial insecurity has been identified as a significant stressor and this, along with part- or full-time work is likely to impact students' socialization and productivity, both of which have consequences as these students prepare for the job market [11,12]. First-generation students are also less likely to perceive the faculty as being as caring and supportive as their peers, which is problematic given that professional mentoring is critical to networking and productivity in higher education [13].

### 1.3. Gender Identity

Women and those with diverse gender identities are also often minoritized in higher education, even if the mechanisms of this process are more complex than with race and socioeconomic background. In some cases, mainstream expectations related to the role of women in caregiving are posited as being incompatible with higher education's expectations related to socialization and productivity [14]. Women in higher education are also often the victims of sexual harassment, where reporting is complicated by unequal power dynamics and systems that are often designed to protect faculty members [15]. There is also evidence to suggest that, at least in some cases, women may work more than men at similar levels of study for less credit, which is again problematic as students prepare to enter the job market [16]. Individuals with minoritized gender and sexual identities are also likely to encounter discrimination in postgraduate education at higher rates than their peers, which increases stress on the student and restricts access to meaningful learning opportunities and environments [17].

### 1.4. Intersectionality

Although research often focuses on one aspect of participants' identities, the reality is that people are not unidimensional. As such, people's various identities (e.g., race, gender, socioeconomic class) often interact with each other in a way that results in unique forms of advantage or discrimination beyond what would be expected based on the individual identities alone. Crenshaw first referred to this concept of intersectionality, which is important to acknowledge in any critical evaluation of systems because the intersections of minoritized identities create unique outcomes that cannot be understood through the exclusive consideration of each identity in isolation [18]. For example, previous research has found that students with various minoritized identities experience their progression through their programs of study differently than their counterparts who do not share these identities [19].

Johnson and Scott reported the experiences of Black women whose degree completion was delayed as well as the factors that may contribute to such delays, with findings supporting that participants' identities as Black women influenced degree completion [20]. More specifically, their participants described having to simultaneously navigate expectations related to being a woman, both in the academy and personally, while also navigating differences in communication and interaction styles related to the intersection of their race and gender identities. Participants reported that this significantly impacted their ability to complete their dissertations and, ultimately, their program of study. Ramirez has also reported some barriers related to students' socialization and academic success in higher education that are unique to the intersection of the Latino/a/x experience in higher education [21,22]. Although the term Latina/o/x may not often be used outside of the North American context, it is retained here out of respect for the original author's decision to use gender-inclusive language in reference to individuals of Latin American descent. Additionally, students' socioeconomic class and background can interact with other minoritized identities to create unique barriers to their success [23,24]. These examples support the claim that students with multiple minoritized identities face unique challenges to their success in higher education that go beyond the individual identities themselves.

These systemic issues of oppression must be acknowledged and addressed to better support doctoral entry, retention, and attainment of all students. When previous research indicates that student identities differentially influence their ability to begin, move through, and complete their programs of study, it is incumbent on institutions to investigate and address these systemic barriers to student success in a meaningful way. This is especially important given that many of the works previously referenced acknowledge the significance of representation in the academy and culturally responsive mentorship practices. In some ways, the best mentors for students with minoritized identities are faculty and staff who share their experiences, but there are two major issues with this solution. First, faculty and staff from historically minoritized backgrounds already report disproportionate and

often uncredited service loads related to this mentorship [25]. Additionally, the issues discussed thus far related to student recruitment, retention, and attainment directly narrow the student-to-faculty pipeline in myriad ways—further disproportionately increasing the load expected of a small group of people relative to the entire workforce at any given institution. In this paper, we examine how previous research highlights promising practices to mitigate these systemic issues through the lens of Gardner's model, which divides the doctoral degree cycle into a series of related phases [26].

*1.5. Phases of Doctoral Development*

Acknowledging that development is the product of both challenge and support, Gardner suggested that institutions can better understand the development of doctoral students by conceptualizing that development as progressing through a series of related phases [26]. She thus suggested conceptualizing three overlapping phases of doctoral student identity development that are defined by their unique challenges and associated supports. Gardner also acknowledges, however, that students can exit their program during any of the three phases if there is a disconnect between their emerging identity, the challenges of the program, and the supports available to them. Broadly speaking, these phases are identified by major hallmarks of the doctoral process: admission, coursework, and candidacy. It is important to note, however, that Gardner's model specifically focuses on the development of students as opposed to reducing their experience to "programmatic turning points or stages of socialization" [26] (p. 9).

The first phase of Gardner's model is characterized as the time between admission and the beginning of coursework. During this phase, students face myriad challenges to their identity as they transition into programs and begin to understand the newfound expectations and demands placed on them in their new role. This phase can have additional challenges for minoritized students, such as students from lower socioeconomic backgrounds who may not be able to attend "optional" socialization activities or first-generation students who are not told the hidden curriculum's implication that such socialization activities (e.g., networking) are critical to developing professional relationships and support networks that will be valuable to their academic and social–emotional success moving forward.

In Gardner's second phase of doctoral student development, the emphasis is on engagement with coursework and transitioning from a consumer of scholarly activity to a producer of such activities. During this phase, students are continuously expected to demonstrate their emerging competence as they progress through the required coursework while also simultaneously beginning to demonstrate their competence in scholarly activities completed in conjunction with their peers and faculty. As mentioned above with regard to phase one, students from minoritized groups may not be told that the emphasis of their development during this phase is on "social and academic integration"—not just completing the courses on their program of study as is the case at the undergraduate level [26] (p. 10).

As students transition out of their coursework, complete required exams, and move into candidacy, the primary focus shifts to the independent completion of their dissertation and finding employment after degree completion. Compared to the first two phases, Gardner recognizes that this phase of development often includes significantly decreased levels of support as students are expected to develop their own independent line of work without the support of external deadlines that are frequently imposed during phases one and two. Additionally, although the timing of this phase is associated with candidacy and work on the dissertation, many demands that emerged during phases one and two continue into this phase as well, such as engagement in collaborative scholarly activities with peers and faculty that may be more immediately pressing or reinforcing but whose work is in conflict with the completion of the dissertation, whose deadlines may appear easier to put off. Again, students from minoritized backgrounds are likely to face additional challenges during this final period of development due to competing agendas and/or

limited support in navigating candidacy and the job market, especially when there are differences in identity or backgrounds between these students and their faculty mentors.

*1.6. Purpose*

In the following sections, several practices are highlighted that may help to address these issues. These practices are organized by the level at which support is available to students to provide a more holistic and inclusive model of doctoral development that explicitly supports minoritized student populations. Practices may be implemented at the level of the institution, individual department, or individual to support minoritized students. Within each level of support, research-based practices are presented that align with at least one phase of Gardner's model so that readers have examples of tools they can use to support minoritized students. In doing so, we propose the use of a SOTL approach in which individual faculty members may consider making changes within their current sphere of influence, working from the bottom-up, to disrupt an educational system that disproportionately reduces attainment for minoritized students. Figure 1 presents an overview of how recommended practices might align with Gardner's model of doctoral student development at the institutional, departmental, and individual levels.

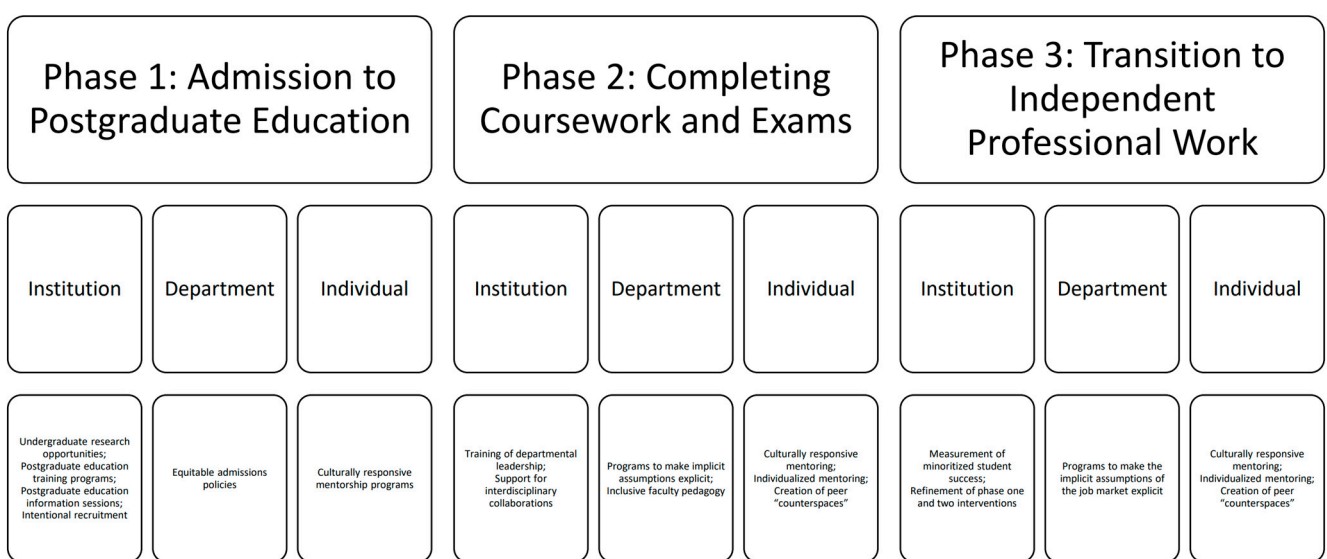

**Figure 1.** Integration of Suggested Supports within Gardner's Developmental Framework.

## 2. Levels of Supports

### 2.1. Institutional Level

While reviews of the literature indicate that institutional change specific to improving doctoral education for students is sparse, institutions serve as important levers of change for improving postgraduate education representation and success [27,28]. Thus, institutions must support and train their administration, support interdisciplinary collaboration, create goals, and document progress to increase minoritized doctoral success [29,30]. Specifically, institutions should focus on creating postgraduate education preparation programs that provide research opportunities and make explicit the hidden curriculum in applying and enrolling in graduate programs. Additionally, institutions should make concerted efforts to recruit and build bridges to minoritized communities and do so in ways that are responsive to minoritized students' needs. Ultimately, departments and individual personnel operate within the guidelines and expectations established at the institutional level, making this level of support a powerful catalyst for or barrier to equitable change in the field.

### 2.1.1. Undergraduate Research Opportunities

Undergraduate research programs are supported by decades of evidence suggesting they support minoritized student populations. The source of these programs varies, with some stemming from federal funding sources and being implemented on university campuses while others are funded and run from universities themselves. Regardless of the funding source, these programs provide undergraduate students with research experiences, often accompanied by faculty and/or graduate student mentorship, career guidance, and financial support. Depending on the approach, these programs can take place over a single semester, multiple years, or even a summer break [28]. Programs can also vary in terms of population served, with some focusing on specific academic disciplines (e.g., STEM fields), years (e.g., seniors), or particular minoritized groups (e.g., first-generation students).

These undergraduate research programs build students' research skills, increase their knowledge of postgraduate education and the associated opportunities, and help with the process of applying to graduate programs [31]. For minoritized students, these programs demystify research, postgraduate education, and the application process by providing workshops, courses, and test preparation for exams that are required for admission to postgraduate education. Through uncovering the hidden curriculum and explicitly guiding students through the process in a supported environment, these programs help build a bridge between undergraduate and graduate education. Although they need significant resources to be effective, previous research on these programs suggests significant positive results in terms of enrolling students into postgraduate education programs, often supporting minoritized communities in particular [32–34]. These programs have also been found to support students through each phase of Gardner's model, suggesting long-term benefits of participation [34].

### 2.1.2. Advertising Postgraduate Education Programs Information

For institutions that may not have student support programs, staff, or research opportunities in place, the recruitment and admission of minoritized students can still be supported by creating smaller graduate education knowledge-building offerings, building relationships with other universities, and demonstrating a positive, anti-racist, and welcoming campus culture. In a study on institutions' diversity officers [28], researchers found that these administrators were able to host postgraduate education information fairs and conferences for minoritized students at their institutions to attend and gain exposure to the idea of postgraduate education and potential career options. These officers also helped support students' attendance at conferences and visits to potential graduate programs, even without the financial support of a fully fledged research program. In this way, institutions can help facilitate a culture that advertises postgraduate education as an opportunity for all students [31]. In sum, institutional offices (whether through the diversity office, admissions, or otherwise) can broadly promote information about postgraduate education and provide resources to their students to engage in recruitment and application activities that minoritized students may otherwise miss.

### 2.1.3. Recruitment

Diversity officers can also intentionally build relationships with minority-serving institutions (MSIs) to facilitate diverse student recruitment as these institutions work with large groups of students with shared, historically minoritized identities [28]. MSIs, which include Historically Black Colleges and Universities (HBCUs), Hispanic-Serving Institutions (HSIs), Tribal Colleges and Universities (TCUs), and Predominantly Black Institutions (PBIs), receive far less funding and have fewer resources to support their minoritized students [35]. So, as MSIs prepare their students for graduation and beyond, it may be helpful for them to partner with non-MSIs who have more resources available to them to help inform minoritized students about research opportunities, provide population-specific postgraduate education information (e.g., application waivers for low-income students), intentionally host students in smaller campus visits, and help ensure a more

diverse population applies to their graduate programs. Importantly, though, bringing minoritized students to campus will not improve diversity unless the campus demonstrates to them that it values diversity [36]. For example, when universities claim to support their minoritized students but officials, faculty, and/or staff act in ways that disregard the lived realities of these students, potential students are likely to see through tokenized statements and efforts to recruit them into spaces that are, in reality, potentially unsafe [5]. All university staff must work to ensure they do not dismiss the feelings of their minoritized students, brush issues under the proverbial rug, or ignore injustices that students face. Minoritized students deserve to be valued, not tokenized, and respected, which will help not only make them want to apply but also support their retention in doctoral programs.

### 2.2. Departmental Level

Departments are also key agents for change given that decisions regarding admissions criteria for graduate programs are largely made at this level. Although universities may have minimum standards for graduate admission, departments often have the latitude to minimize or eliminate requirements such as standardized test scores, a decision that would support the admission and enrollment of minoritized students [28,31]. Requiring standardized test scores for admission may be especially problematic for minoritized student recruitment since a recent meta-analysis showed that standardized test scores are not predictive of outcomes such as student retention or completion of doctoral programs [37]. Through policy and practice, faculty can create either positive or negative environments for their students. In this section, we suggest practices that faculty can adopt to create departmental cultures that encourage and sustain learning environments that are inclusive and supportive of minoritized students. As departments work to change their internal operations to be more equitable in their support of all students, it would be helpful for them to consider sharing those efforts in the spirit of the SOTL to foster similar changes within their institution and in comparable departments at other institutions [3].

### 2.2.1. Coursework

Faculty can intentionally create graduate-level learning opportunities—formally or informally—to demystify academia and acknowledge societal inequities that pertain to PhD students. To illustrate, Moore and colleagues described a seminar provided by senior Black faculty to support junior Black faculty and doctoral students [38]. Within the seminar, the senior faculty described aspects of academia not commonly reported in textbooks, such as developing a research agenda, describing the tenure and promotion process, and navigating departmental politics. Importantly, the faculty leading the seminar took it upon themselves to create the seminar in an act of "paying it forward" based on their own experiences with others supporting them when they were doctoral students and junior faculty. Similar courses and learning opportunities can potentially address different milestones or phases within the doctoral degree process and may be offered either as a required course or as a supplemental informal offering. For example, Renbarger and colleagues reported on an informal seminar focused on uncovering the hidden curriculum of applying for academic positions [39]. Departments could also consider creating coaching groups for students so that they can learn from mentors from multiple universities [40]. As such, departments can tailor their formal and informal offerings to best meet the needs of their particular student groups and equitably support minoritized students throughout their enrollment.

### 2.2.2. Faculty Pedagogy

Even outside of specific coursework, faculty can support PhD students through how they teach. In a study on a higher education and student affairs graduate program, the graduate students of color noted that effective faculty engaged in three pedagogical strategies that supported students' success [9]. The first strategy involved faculty intentionally incorporating social justice in their teaching, such as through bringing up race and racism when students avoided the topic, finding resources to support race consciousness, and

making assignments relevant to their lives and future work. These faculty also encouraged discussions in class and on online platforms to allow classmates to share their perspectives and create a community where students could empathize and sympathize with others, particularly around identity and social justice. Finally, faculty encouraged collaboration inside and outside of the classroom that allowed students to continue their critical conversations, enjoy camaraderie with others, and build networks with other students of color [9].

### 2.3. Individual Level

Changes at the institutional and departmental levels can take a long time to manifest due to the need for unofficial and official buy-in from multiple parties. Although one of the most important groups for improving equity includes faculty members, we argue that all members of an institutional community can support minoritized students' success through one of the most established evidence-based practices: mentorship [41]. Although individuals can, and often do, work in isolation to provide mentorship to minoritized students, systemic change is more likely to occur through an embrace of a SOTL framework that involves generating change across individuals, departments, and broader institutions [3].

### 2.3.1. Faculty

Faculty mentoring has long been heralded as a best practice for improving student outcomes [42,43]. More recently, researchers have called for the need for culturally responsive mentoring, particularly for students of color. Similar to the activities in the pedagogy section, this mentoring includes "cultivating a critical consciousness, decoding the hidden curriculum, and developing cultural awareness as mentors and mentees" [44] (p. 399). While some researchers state that culturally responsive mentoring includes more one-way mentorships from faculty to student, these relationships may also be reciprocal, two-way mentoring relationships [44,45]. In either case, both mentors and mentees should bring their authentic selves to the table, honoring their racial, cultural, and gender identities and life experiences even within institutional settings that do not honor these backgrounds. Mentors thus play an important role in supporting students' abilities to navigate these unwelcome environments, first building trust and compatibility through sharing their own experiences [44,45]. These advisors must provide positive doctoral coursework advising as well, although this alone will not retain students in their program [46]. Once the relationship is built, mentors then socialize students to academia, such as by helping them publish academic work, involving students in networking opportunities, and answering questions about non-academic career paths. Mentors should also humanize themselves and mentor beyond academics where appropriate as this personal mentoring mostly occurs for privileged rather than minoritized groups [47].

These supports should serve to uncover the hidden curriculum and provide opportunities that align with the students' interests and values, acknowledging that no group exists as a monolith [44,45]. When relevant, mentors must support mentees' search for careers outside of academia as students may or may not see how racism, sexism, and multiple marginalization negatively impact faculty well-being. Mentors can help uncover these realities, provide tips for navigating academic job applications, and connect students to scholars in other career paths that better align with the goals and identities of their particular students [44]. These discussions must take place multiple times throughout the doctoral journey as research has found that students' desire for academic careers decreased over the course of their doctoral studies, with men choosing a career in academia at higher rates than women [48]. Examples such as mentoring for Black students and alumni or feminist mentoring can support faculty uptake of culturally responsive mentorship practices [49,50]. Faculty engaging as mentors can vary, from advisors, to others in the graduate program, or those across the institution who share similar identities [51].

### 2.3.2. Administrators and Staff

Of course, mentorship of doctoral students does not and should not fall squarely on the shoulders of faculty members. Minoritized students may need a network of support from players across the institution [51]. In what has been described as a "cascading mentorship" model, more senior members of graduate programs and laboratories can support doctoral students' skill development [52]. In examining biological science PhD students, researchers found that senior graduate students' and postdoctoral researchers' participation in laboratory discussions positively related to doctoral students' research skills far beyond that of faculty members' specific mentoring practices [52]. Other relationships, such as those with program directors, administrators, and institutional leaders, may support minoritized students academically or emotionally and thus support retention by providing encouragement and resources [31,51]. Mentors should encourage a multi-layered approach that encourages students to find multiple mentors who can offer support in various areas, particularly when the mentors do not have similar racial, gender, or cultural background similarity [44].

### 2.3.3. Peers

Given the small numbers of minoritized faculty and staff at universities compared to the number of students, students often mentor each other. These mentorship groups may be formal or informal and have been given a variety of labels depending on their makeup and purpose. For example, Black doctoral women created sister (or "sista") circles for navigating the doctoral journey at predominantly white institutions using Black feminist mentoring practices [53]. These groups often include doctoral students at various parts of their journey and do not maintain systems of hierarchy between the members. Because students must often mask aspects of their identities in their departments or institutions, these counterspaces create a space of belonging by allowing doctoral students to be their authentic selves and can also encourage research collaborations [53]. As an illustration, doctoral students created a feminist research group that emphasized collaboration over competition and demonstrated authentic care toward each other [54]. Peers can impart important information beyond what faculty can, such as how to navigate politics, and provide more frequent encouragement and support [51]. Depending on the peer, they may also be able to support students' job searches at the end of their doctoral program as well, although many peers may provide misinformation or have no more information than the job seeker [55]. To aid in the connection of students to each other, faculty can facilitate classroom counterspaces for their students within courses that feature the experiences of Black and Brown students (e.g., ethnic studies, indigenous methodologies) to co-construct the course, amplify their voices, and provide a space for vulnerability [56].

## 3. Conclusions and Future Directions

We advocate for a comprehensive and evidence-based approach to address systemic barriers in graduate education and offer practical strategies to promote equity and support the success of all students, particularly those from minoritized backgrounds. Disparities in access and success within graduate education can be understood through the context of students' minoritized identities as well as the intersection of those identities. Existing systems perpetuate these disparities, but evaluating these systems within Gardner's framework of doctoral student development and using an approach similar to that of the SOTL may lead to more equitable solutions at all levels of the university: institutional, departmental, and individual faculty and staff. Challenges faced by historically minoritized students may have a similar root cause, such as racism, at each level, but the student experience and topography of the challenge may look different. For example, a student experiencing racism at an institutional level may be deterred from entering programs in a chosen field while departmental-level racism may assign this student to a weaker or less productive mentor, harming the student's chances of developing a successful career in academia. Either

option leads to reduced outcomes for the student through opportunity costs; however, to observers, the symptoms of the harm appear very different.

By using Gardner's Model within a SOTL framework, institutions, departments, and individual scholars can identify solutions at doctoral entry, retention, and attainment for all students and promising practices can be implemented at each level of influence within the university (e.g., institutional, departmental, individual). Some promising practices identified by previous research include undergraduate research opportunities, intentional recruitment practices, pedagogy reflective of the students' needs, and culturally responsive mentoring. Individuals with administrative appointments who are dedicated to these efforts may also magnify the supports offered by individual faculty members and departments.

However, institutions must be mindful of how implementing these practices may affect already-minoritized faculty members, such as women and faculty of color who already bear a larger burden of service compared to their white male counterparts [45]. When considering how other institutional actors, such as postdocs, can provide support to doctoral students, institutions must provide training in mentorship, especially culturally responsive mentorship practices, and adequately pay them given the benefits they provide students [52]. Additionally, all institutional actors need proper training on how their actions uphold structural inequality. Institutions as a whole must change or risk continuing to hurt their graduate students.

Future researchers should endeavor to continue documenting harms experienced by minoritized students in addition to reporting the results of initiatives designed to mitigate or prevent such harms at various levels of practice. Such publication and dissemination are consistent with a SOTL framework and are critical to fostering systemic change in the field so that minoritized students are equitably supported at each stage of their doctoral journey from recruitment through graduation. Additionally, the use of a SOTL framework to document and drive change in higher education helps to validate the lived experiences of minoritized students while also supporting equitable changes to support their success.

**Funding:** This research received no external funding.

**Institutional Review Board Statement:** Not applicable.

**Informed Consent Statement:** Not applicable.

**Data Availability Statement:** No new data were created or analyzed in this study. Data sharing is not applicable to this article.

**Conflicts of Interest:** The authors declare no conflicts of interest.

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
