# Peer review of "Improving Equitable Access to Graduate Education by Reducing Barriers to Minoritized Student Success"

_education, doi:10.3390/educsci14030298_

Round 1

Reviewer 1 Report

Comments and Suggestions for Authors

This review of the literature constitutes a potentially valuable resource for academics and practitioners interested in making access to and success in and beyond doctoral level study more equitable. The paper appears to be well researched and is well written. My only two reservations in recommending this paper for publication are (1) by covering racial, socioeconomic and sex inequalities, this paper arguably is not able to do full justice to the distinctive challenges faced by those who are minoritised in these three quite different respects, and (2) as far as I can ascertain, the initiatives described and advocated by the authors seem to be of the sounds-like-a-good-idea variety but without solid research evidence to demonstrate their efficacy. With regard to this latter point, I would have expected to see more in the way of evidence evaluating the effectiveness of the various interventions discussed.

Author Response

Thank you for the opportunity to complete minor revisions to this manuscript. Please see the attachment for our responses to your comments on how to strengthen this work. 

Reviewer 2 Report

Comments and Suggestions for Authors

Thank you for the opportunity to review the paper. It is well written, well structured and of interest to many involved in higher education and equality issues. 

I think with some minor revisions, the paper could be further improved in approach and clarity for readers. 

It is clearly written from a North American perspective. That is fine, and will still be of relevance to a wider readership, but it would be better to make that clear. There are some concepts that do not translate to other contexts - p108 ref to Latino/a/x experiences, p5 Majors and, in general, Graduate Schools and Minority Serving Institutions. 

Leading on from that, there are terms/concepts that require further explanation for an international or unfamiliar audience. I would suggest you need to define: hidden curriculum (line 147), GRE (line 218) and Sankofa mentoring (line 352). 

I also think it would be useful to reflect Gardiner's model within the discussion of activities hat can be undertaken. This would draw together the conceptual basis of the paper with the wider narrative. 

Finally, I think the experiences of trans, non-binary and gender diverse people could be included within section 1.3. There is research evidence to suggest they also have a poorer experience within HE as a minoritised group. 

Comments on the Quality of English Language

The paper is very readable. I suggest, however, reworking paragraph 110 - 110 as I found it a little unclear in its present form. 

Author Response

(The authors gave the same response as above.)
